# Identity leadership and adherence to COVID-19 safety guidance in hospital settings

**Kayleigh Smith**, **Anne Templeton***

Department of Psychology, University of Edinburgh, Edinburgh, United Kingdom

* a.templeton@ed.ac.uk

## Abstract

COVID-19 presents unique and complex challenges to the Scottish National Health Service (NHS). As COVID-19 preventative measures are effective at reducing disease spread, promoting staff adherence in high-risk workplaces is vital. The present research explored the role of identity leadership on (a) staff's appraisals of leadership and (b) staff's adherence to and attitudes towards COVID-19 guidance. Semi-structured interviews ($N = 25$) were conducted with NHS staff across two Scottish hospitals. Using Reflexive Thematic Analysis, two over-arching themes were generated: leadership presence and approachable leadership who act on group concerns, where both created positive appraisals of leadership and were seen to facilitate adherence. Guidance from present leaders was perceived as both practical and applicable. Approachable leaders were viewed to facilitate information sharing, clarify guidance, and allow staff to raise concerns. Leaders who were seen to act on group concerns provided resources or updated guidance to promote adherence. The present study provides theoretical and practical advancements to (a) expand the known role of identity leadership in promoting safety in workplaces and (b) facilitate routes for adherence to safety guidance beyond COVID-19.

## Introduction

COVID-19 continues to present unique and complex challenges to health services. The Scottish National Health Service (NHS) has struggled with staff shortages [1] and increasing demands due to the backlogs accumulated throughout the pandemic response [2]. Hospitals may facilitate high COVID-19 transmission through contact between staff and infected patients, thus promoting staff adherence to COVID-19 safety measures (e.g., wearing of facemasks and physical distancing) [3] is key to preventing the spread of infectious diseases. One potential avenue to promote adherence is through the role of effective identity leadership [4–6].

Existing guidance within the NHS proposes a nine-dimensional model for effective hospital leadership [7]. These dimensions overlap theoretically with principles of positive identity leadership, including facilitating feelings of a shared goal and purpose between staff [8], sharing the key vision of the NHS with team members and acting upon that vision [9], and actively engaging with team members to promote team involvement as opposed to autocratic leadership styles [5]. However, no known research to date has explored the role and effectiveness of

**Data Availability Statement:** All transcripts are available on the Open Science Framework and can be found at https://osf.io/ajumr.

**Funding:** This research is funded by the Chief Nursing Officer Directorate of Scottish Government

for the project 'Identifying barriers and avenues to safe behaviours in high-risk workplaces'. The funders had no role in study design, data collection and analysis, decision to publish, or preparation of the manuscript.

**Competing interests:** The authors have declared that no competing interests exist.

identity leadership within national health services. Further, research is yet to explore whether principles of identity leadership can encourage safe behaviour to prevent the spread of infectious diseases within healthcare settings.

The present study aims to 1) evaluate whether leaders who are perceived positively by staff are discussed in relation to identity leadership, and 2) explore whether identity leadership is related to staff's reported adherence to, or attitudes towards, COVID-19 safety guidance within hospital settings.

## Social identity approach

The social identity approach [10]–comprising of social identity theory [11] and self-categorisation theory [12]–provides the theoretical background to understand the role of leadership adopting group processes to promote safe behaviour. Social identity theory posits that everyone belongs to multiple social groups, where one's self-concept can range from their *personal* identity (their idiosyncratic attributes) to their *social* identity as a member of any particular social group. These group memberships provide information about the social world, including what values and behaviours are normative for each social group. Self-categorisation theory highlights how individuals shift from their personal to social identity through depersonalisation, and begin to adopt the norms, values, goals, and behaviours of the social group through self-stereotyping.

The social identity approach to leadership argues that leadership can influence and lead groups *through* the group as opposed to *at* the group [5]. Identity leaders can demonstrate their group membership and use this to illustrate which behaviours are appropriate within that social group. For example, successful political leaders throughout the COVID-19 pandemic adopted a group approach to messaging to encourage adherence to COVID-19 guidance [4]. Thus, the social identity approach to leadership demonstrates the potential for leaders to adopt group processes to encourage safe behaviour.

## Dimensions of identity leadership

The social identity approach to leadership can expand our knowledge of how to encourage engagement in safe behaviours within hospitals. Steffens et al. [9] propose a four-dimensional approach to identity leadership: identity prototypicality, identity impresarioship, identity entrepreneurship, and identity advancement. These dimensions highlight how leaders are seen to represent ideal membership of the group (prototypicality), how leaders can provide opportunities for group members to show their group membership (impresarioship), how leaders can craft a sense of 'us' and togetherness (entrepreneurship), and how leaders can act for the group and on behalf of the group (advancement). Initial discussions from the wider project with select NHS staff in Summer 2021 highlighted that frontline staff felt distant from management and that they had limited communication and engagement from leadership (see a summary report from the larger project on OSF; https://osf.io/ajumr). Consequently, staff felt a lack of support from leadership which presented challenges to their adherence to the COVID-19 safety guidance. Since staff raised questions about leaderships' representativeness of the group, and discussed limited communication and engagement from leadership, these conversations highlighted the particular relevance of identity prototypicality and advancement for understanding staff adherence to guidance.

**Identity prototypicality.** Identity prototypicality refers to when leaders represent what it means to be a member of the group [9, 13]. Specifically, it refers to not only how leaders represent the 'average' group member, but also how leaders represent what it means to be an 'ideal' group member [14, 15]. This appraisal of prototypicality comes from group members assessing

whether leaders represent the values, behaviours, and goals that are normative of the social group [13].

Leaders who demonstrate prototypicality are perceived more positively than non-prototypical leaders. For example, leaders who demonstrate identity prototypicality tend to be rated as more effective [15, 16]. Ratings of leadership effectiveness were also found to drop less following leadership failure when leaders were perceived as prototypical [17, 18]. In addition, prototypical leaders facilitate greater support from followers to achieve group goals [19], including group members demonstrating greater endorsement and support for unfair leadership when these leaders are viewed as prototypical [20]. Finally, leaders perceived as prototypical group members facilitate greater feelings of trust from group members compared to non-prototypical leaders [19, 21]. This association was recently explored within an NHS hospital, where increased perceptions of leadership prototypicality were linked to increased reported trust in leaders to appropriately handle the COVID-19 situation [22].

Identity prototypicality not only contributes to positive appraisals of leadership but can also encourage followership and safe behaviour. For example, research exploring predictors of COVID-19 adherence has found significant associations between trust in leadership, or trust in government, and people's willingness to adhere to COVID-19 restrictions imposed by government [23–25], where this trust in leadership can be facilitated by leaders demonstrating group prototypicality [19]. Thus, prototypical leadership within hospitals could be key to promoting staff adherence to safety guidance. Finally, the social identity approach to leadership highlights that leaders can be influential in guiding group behaviour through demonstrating group prototypical behaviours, which followers can look to when inferring how they are supposed to act [13]. Within the current context, leadership engaging in COVID-19 safety guidance could indicate both (a) leadership group membership and (b) how staff are supposed to act (by adhering to guidance). Thus, leaders who demonstrate identity prototypicality facilitate both positive appraisals of their leadership and followers' engagement in behaviours important to the group.

**Identity advancement.** Identity advancement refers to leaders being seen to be acting on behalf of and standing for the group [5, 9]. Identity advancement involves leadership promoting the values and goals of the group, as opposed to promoting outgroup goals and values [26], where leaders are seen to be getting things done for the benefit of the group as opposed to acting upon their self-interests [9].

Leaders who demonstrate identity advancement have also been found to be perceived more positively compared to leaders who do not. For example, leaders seen to be prioritising the group's goals above their own (e.g., by sacrificing their time to advance the group) are rated as more effective by group members [27]. Similarly, leaders seen to be displaying group-oriented attitudes and behaviours are endorsed more than leaders who display self-orientated attitudes and behaviours [28]. Additionally, leaders who are perceived to be acting upon the goals and interests of the group are rated as more authentic leaders [29]. Finally, within the validation of identity leadership scales, significant associations were found between identity advancement and followers' feelings of trust in leadership [21].

Similar to identity prototypicality, identity advancement results in both positive appraisals of leadership and routes to promoting followership and safe behaviour. First, as discussed with identity prototypicality, identity advancing leaders can facilitate greater feelings of trust from followers [21] which is associated with promoting adherence to COVID-19 safety guidance [24]. Further, leaders who promote the interests of the group above their own facilitate group members engaging in supportive behaviours towards that leader, including voting for these leaders [29] and supporting leaders' policies [26].

Both identity prototypicality and identity advancement contribute to positive appraisals of leadership and facilitate followership and engagement in behaviours important to the group. Thus, within the NHS, identity leadership could be relevant to both positive evaluations of hospital leaders and increased adherence to COVID-19 safety measures.

## Present study

The present study aims to explore the role of identity leadership on perceptions of hospital leadership and staff adherence to COVID-19 safety measures. Using Reflexive Thematic Analysis [30], the present study aims to address the following research questions: 1) Is leadership within the hospitals discussed in line with the phenomena related to identity prototypicality and identity advancement? 2) Is identity leadership (or lack thereof) related to staff's self-reported adherence to or attitudes towards the COVID-19 safety measures?

## Methods

### Ethics and open science statement

The present study was approved by the University of Edinburgh PPLS Ethics Committee (reference 385-2021/4). The research was pre-registered on the Open Science Framework (https://osf.io/ajumr).

### Participants

NHS staff from two Scottish hospitals in different health boards participated in online interviews between October 11[th] and December 20[th] 2021. When data collection began, 87% of the population in Scotland aged 18+ had received two doses of the COVID-19 vaccinations and overall positive COVID-19 cases were down, but the government were concerned about increased infection rates as winter approached. Participants were recruited via advertisements on staff intranets, emails to staff, flyers distributed across both hospitals, alongside snowball sampling methods. Stopping criteria were set for (a) when data was saturated and (b) when participant availability became limited due to increasing demands of the pandemic (e.g., the increasing prevalence of the Omicron variant). Staff from various roles in the hospitals (e.g., nurses, porters, administration) were invited to encourage a broad range of viewpoints, however participant demographic information was not stored to further protect anonymity. A total of 25 interviews were conducted ($n$ = 16 hospital A; $n$ = 9 hospital B), where nine were selected for in-depth analysis due to providing particularly in-depth, rich data that was representative of all interviews. Codes were developed from these initial nine interviews before being applied across the remaining 16 transcripts. Participants received a £10 e-voucher for their participation.

### Interview procedure

Interviews were conducted via Microsoft Teams or phone and recorded via Microsoft Teams in both instances. Interviews ranged from 14 minutes to 49 minutes (mean interview time = 36 minutes, total length of all interviews = 902 minutes). Interview times often varied due to the demands of the pandemic, such as staff being delayed from working longer than anticipated. All participants provided written and verbal informed consent prior to the interview. Interviews recordings and participant information were stored on a secure OneDrive server only accessible to the research team.

Individual semi-structured interviews were conducted to (a) allow the researcher to guide the conversation to areas of interest but with flexibility for follow-up discussions depending on

responses from participants, (b) allow participants to raise important topics and steer the conversation [31], and (c) allow the researchers to allocate priority questions for the interviews in case participants did not have very long to participate.

Participants were asked about their relationship with their line managers, what they felt contributed to their relationship with leadership, their views on whether conversations between the team and leadership were two-way, and how they would feel if leaders were seen to not adhere to the safety measures (see S1 File for the full interview schedule). The questions asked about staff teams and management but allowed staff to create their own definitions of who was part of their team/group given variability in ward and staff structures. Participants were debriefed verbally after the interview, with this being emailed to participants upon request.

Interviews were anonymously transcribed, including the removal of any identifying information to protect anonymity (e.g., participant name, the name of a colleague, the specific ward the participant is employed in), and each participant was given anonymous identification initials.

### Analytic process and approach

Reflexive Thematic Analysis [30] was used following the six-phase structure set by Braun and Clarke [32]. The first author became familiar with the data through verbatim transcription of the interviews alongside actively reading through the transcripts and taking initial notes on areas of interest. Initial coding was conducted including codes that were both semantic (the surface level explicit meanings present in the data) and latent (consideration of the underlying meaning beyond the data). Coding was then discussed and revised before being collated and organised into themes and subthemes. Selected themes and subthemes were then checked for accuracy against the data before being allocated names and definitions.

The analysis was conducted with a mix of deductive and theory-driven approaches [33]. Specific research questions were used and themes were generated with an awareness of the prior literature on identity prototypicality, identity advancement, and their influence on group behaviour. However, we were open to identifying new codes or themes that were not initially expected based on prior literature. The analysis took a pragmatist ontological approach, where we assumed that there is a real-world/reality to be explored [34]. However, this was coupled with a constructionist epistemological approach [35], where we did not aim to explore objectively whether hospital leadership was (in)effective. Instead, we explored staff's subjective perceptions of leadership and took an experiential approach [36] to staff's experiences and perspectives. Throughout the analysis, we acknowledged the challenging COVID-19 context and the increased pressures experienced by NHS staff. Finally, we also took a critical orientation [37] whereby we were reflective and acknowledged the limitations of what staff's experiences could tell us (e.g., where staff's personal experiences with leaders, such as feelings of animosity, could influence their reported experiences).

**Credibility strategies.** First, a broad variety of viewpoints were evaluated by including staff from different departments across two hospitals, including those in leadership positions. Second, peer debriefing was adopted wherein a staff working group associated with this project evaluated whether our findings were consistent with their understanding, alongside consultations with hospital management. Third, we adopted reflexivity by looking back on our theoretical assumptions and our positionality towards healthcare workers and their experiences throughout the pandemic.

### Analysis

Across the interviews, staff highlighted that leaders who were approachable, engaged, and acted for the group were viewed most positively and facilitated adherence to guidance.

**Table 1. Summary of over-arching themes, themes, and descriptions.**

| Over-arching theme | Theme | Description |
|---|---|---|
| Leadership presence and engagement demonstrating ingroup/outgroup membership and prototypicality | Present leaders demonstrating prototypical behaviours signals ingroup membership | Leaders being present and engaging in prototypical staff tasks facilitates perception of them as ingroup members |
| | Non-present leadership lacking shared views | Non-presence of leadership (being in their 'ivory tower') creates contrasting expectations and clashes between frontline staff and leadership |
| | Leaders who are present having shared experiences with frontline staff | Leaders who have the same experiences as frontline staff facilitate (a) feelings of leadership being ingroup or prototypical group members and (b) adherence via guidance being applicable and practical |
| Approachable and engaged leaders who act for the group | Approachable leadership | Approachable leaders facilitate feelings of support, clarify the guidance enabling staff to adhere, and enable two-way conversations to allow the co-production of guidance built off frontline staff's experiences |
| | Engaged leadership who act upon staff's concerns | Leaders who act on the concerns of staff facilitate support, faith, and trust that leaders will continue to act on behalf of the group (identity advancement), where leaders acting on the concerns of the group provides opportunities for improvement, with this facilitating adherence |

Specifically, the presence of leadership on the ward and understanding the difficulties staff faced was important in staff seeing leaders as prototypical group members and feeling supported to follow the guidance. See Table 1 for a summary of over-arching themes and themes.

## Leadership presence and engagement demonstrating ingroup membership and prototypicality

**Present leadership demonstrating prototypical group behaviours.** Staff positively appraised leaders who were present and engaged in the same prototypical tasks as frontline staff:

> Our immediate line manager has been fairly visible. It's kind of been doing what we've been doing. There's not been any reluctance to go and get stuff done
>
> (Extract 1, RE, hospital B)

After being asked about whether RE (extract 1) felt their relationship with leadership was a feeling of being 'in it together', RE highlights how their immediate line manager was seen to be on an equal level as themselves by doing the same prototypical tasks as frontline staff, where this leader demonstrated a willingness to share these responsibilities. This lack of reluctance to engage in frontline tasks is viewed by RE to demonstrate that the leader was one of the team. Within this conceptualisation, leaders who are present and perform the same prototypical tasks as staff are viewed more positively than leaders who are seen as absent or not engaging in the same work as staff.

This is also discussed by OC (extract 2) when asked about their own relationship with leadership at line manager and senior management levels, but OC further shows how leadership engaging in prototypical staff tasks was a signal of group membership. OC highlighted their own positive experience before moving on to discuss their perceptions of the relationship between frontline staff and leadership on other wards who were not present:

> [Staff on other wards] mistrust was maybe 'I'm feeling like a lamb to slaughter' wasn't really like directed at their line management say but sort of further up the chain or like hospital

management, potentially even government. Yeah I think from what I understand most of their line management were like sticking right in as well and doing whatever they could too and were doing as much clinical work as they possibly could

(Extract 2, OC, hospital A)

OC discusses feelings of mistrust and abandonment being created by those higher in the leadership hierarchy, i.e., leaders who are non-present and distant. In contrast, OC views line managers who are seen to fully engage ('sticking right in') in frontline tasks as a positive and distinct group from upper management. By comparing how line managers and upper management engaged with staff, OC removes line managers from being responsible for feelings of mistrust towards leadership or abandonment from leadership. Instead, the presence and performance of shared tasks by leadership are antecedents to feeling part of a group with the leader, because they signal that those leaders are equal.

**Non-present leadership lacking shared views.**   In contrast to present leaders being seen as prototypical, more distant leaders were seen as not sharing the same views as frontline staff and therefore lacking prototypicality:

I think it very much felt like those on the ground versus those in the ivory towers. . . we're the ones on the front line with PPE that didn't work and other people [leaders] were, you know, complaining that people were going off sick with covid or similar

(Extract 3, AE, hospital B)

In extract 3, AE creates a category of non-present leadership. First, AE crafts a categorisation of non-present leadership as 'other' through describing them as being in the "ivory towers", symbolising these leaders as further up the hierarchy with a lack of awareness of the issues faced by people lower in this hierarchy. Second, AE further categorises the group membership of these leaders by describing non-frontline staff as "other people", attributing a category definition to those who are not present as outgroup compared to the ingroup of present frontline staff. In this conceptualisation, leaders are viewed as being distant not only physically, but distant from the experiences of frontline staff. The lack of understanding of the frontline staff experience AE perceives from leadership (through complaining about staff going off sick) is attributed to this distance and separation between frontline staff and leadership.

By being perceived as non-present, and therefore lacking an understanding of the challenges faced by frontline staff, any relational connection between hospital leadership and frontline staff is seen to be lacking. Thus, there are limited antecedents to shared social identity through leadership distance. Further, given staff feel there are barriers to adhering, such as a lack of working PPE, leaders who are not present and therefore lack an understanding of the views of frontline staff are seen as unlikely to update guidance to remove these barriers.

SA (extract 4) further exemplifies viewing non-present leaders as lacking shared views with frontline staff when discussing the ability to talk to leadership about the issues staff are facing:

With higher management it got a bit trickier just because obviously they are all working from home and not actually in the department and, you know, not actually seeing what we're seeing on a daily basis so I think at times there was a bit of butting heads shall we say [. . .] we're not, you know, complaining for the sake of complaining. We are complaining

because we cannot do our jobs [. . .] when it reaches a point where you can't actually do your job safely, that's when you know we've really had to fight this

(Extract 4, SA, hospital A)

Leaders who are not present on the floor are seen as unable to recognise the challenges faced by frontline staff, where contrasting views of what challenges staff are facing results in clashes between frontline staff and leadership, contributing to an overall negative perception of leadership. Recognising others as ingroup members is, in part, attributed to seeing others as sharing the same goals and values as us [14]. In this example, non-present leaders are seen to have different viewpoints and expectations to frontline staff. This, in turn, contributes to a lack of shared social identity between frontline staff and leadership and limits the ability of staff to adhere to safety guidance by leaders not being there to listen and act.

**Present leadership having shared experiences with frontline staff.** Non-present leadership also contributed to views of leaders having different experiences of the pandemic response:

I don't think the challenges there [in the leader's office block] is anything like they are in the ward setting in terms of space and distancing and numbers and even things like silly things that don't really seem to affect him [leader] over there is like the temperature when you're wearing the PPE <u>all</u> day every day. I mean obviously they are wearing masks as well but they're not wearing gloves and gowns all day every day in the office block. And so I think it is hard for them to kind of really appreciate exactly how tough it is on the ward

(Extract 5, NO, hospital B)

NO (extract 5) provides an example where a leader who worked within a separate environment to frontline staff was seen to have different experiences of the PPE challenges faced on the wards. Prior literature demonstrates the role of shared experience in recognising others as part of the same social group [38]. Here, recognising a lack of shared environment and therefore lack of shared experience between frontline staff and leadership renders the leaders in the office block as "other" to frontline staff on the ward.

Perceived lack of shared experience between leaders and frontline staff was also seen to contribute to the ability for staff to adhere to safety guidance. AE (extract 6) demonstrates this in the following example, where they were asked to further discuss why they felt that emails communicating guidance were not effective, where AE was unaware of who these emails were coming from:

I think a lot of people that write those emails have never actually done the job, or they say 'you should do this for this situation' but the ward doesn't have a sink, or the room that you're in doesn't have a sink or doesn't have an alcohol gel dispenser in it [. . .] it's just that the guidance that comes down from wherever it comes from doesn't seem to have ever, people don't seem to have applied it or tried to apply it [. . .] It's often very irrelevant from what's going on on the ground

(Extract 6, AE, hospital B)

AE provides an example where guidance was felt to be challenging to adhere to because the work environment did not have the resources required to follow the guidance (for example, a lack of hand sanitiser or environmental restrictions), leading to the guidance being viewed as impractical. AE attributes this impractical guidance to non-present leadership who were

unable to understand or appreciate the working environment due to a lack of shared experience with frontline staff. AE thus demonstrates a view that in order to facilitate staff adherence to safety guidance, leaders need to be present and have a shared experience as to provide guidance that is practical to adhere to within the constraints of the hospital or specific ward environment.

Notably absent from AE's comments on leadership is feeling part of the same group as them. In line with the prior literature that a sense of common experience can lead to shared social identity among the people facing the situation, AE's comment on distant leadership lacking shared experiences demonstrates how the antecedents to feeling part of a group with leaders are missing when leadership are physically and psychologically separate. Moreover, they show the subsequent implications this can have for non-adherence to safety guidance. In contrast, AE (extract 7) positively appraises the suggestion of having a single, present person responsible for translating and providing safety guidance (which was the situation in both case study hospitals):

> I think the best thing about that [having a single point of contact for covid guidance] is that person translates what is the kind of senior management policy into something that is actually do-able. And it needs to be someone that's close to the ward that's doing that as well

(Extract 7, AE, hospital B)

Importantly, AE highlights that for guidance to be practical, applicable, and thus adhered to, they feel it needs to come from someone who is present on the floor and who has engaged in shared experiences with frontline staff. As presence and engagement with frontline tasks have been viewed as demonstrating leadership prototypicality (i.e., sharing the experience and acting as the group should), there is a suggestion that guidance should come from those viewed as prototypical to ensure the guidance is practical and able to be adhered to. Further, AE highlights a viewpoint that a present single point of contact for guidance is perceived as someone who acts on behalf of the group. Thus, this suggestion of a single leader in charge of guidance communication, who could ensure practicality and applicability via their presence, could promote both positive appraisals of these leaders as identity advancing, alongside facilitating the ability for staff to adhere to COVID-19 safety guidance by providing practical guidance that can be adhered to.

### Approachable and engaged leadership who act for the group

**Approachable leadership.** Non-approachable leadership were negatively appraised, where staff believed that these leaders left their colleagues feeling abandoned and dismissed:

> I think for us it was a two-way conversation [between staff and leadership] and we felt we were able to say what was right and what was wrong and voice our concerns, but a lot of people, my colleagues, others, didn't have that at all, felt pretty let down [. . .] staff were in tears, not just nurses but doctors too, because they were so scared and basically being made to feel like lambs in slaughter

(Extract 8, OC, hospital A)

OC (extract 8) highlights contrasting staff experiences of approachable leadership within the hospital. They report positive experiences of approachable leadership where they felt able to voice any concerns they had but acknowledged that this was not a universal experience, where non-approachable leadership was viewed as an experience shared by many. This lack of

approachable leadership was viewed by OC to leave staff feeling highly dismissed and abandoned.

Beyond facilitating feelings of support, a recurring theme was that approachable leadership was also key to facilitating feelings of group membership between leaders and frontline staff, alongside providing the information required to allow staff to adhere to COVID-19 safety guidance:

> The relationship's great. Actually I have to say within my experience of the relationship between staff here and management here is very good, they're very approachable. I don't think everybody will tell you the same thing, but my relationship is that they're very approachable and that, you know, they treat you almost as an equal as it were, so you get as much information as they've got
>
> (Extract 9, IR, hospital A)

IR's experience of approachable leadership (extract 9) further demonstrates the link between approachability and both positive appraisals of hospital leadership and the ability for staff to adhere to the guidance. The sense of solidarity described by IR between leaders and frontline staff was seen to create an avenue for sharing information, putting leaders and staff on equal footing and creating a sense of staff and leadership being in it together. Approachability was seen to facilitate this two-way interaction between leadership and frontline staff, which in turn allowed staff to be provided with the same level of information as leadership. Having up-to-date information was recognised by other participants as key to facilitating adherence to current guidance; for example IR (hospital B) highlighted that not having the most up-to-date information meant that being able to "keep on top of what we were meant to be doing was quite challenging".

Further, approachable leaders were viewed positively, particularly for their availability to provide clarity to guidance to help staff, as discussed by IR (extract 10) when asked about their conversations with leadership on the ward:

> It's quite an open kind of conversation that you can ask questions if you're unsure about anything, everybody is really approachable about it
>
> (Extract 10, IR, hospital B)

IR assumed that this leader was consistently happy to help the group by being available to clarify the guidance, where this line of communication was seen as regularly available when required due to leadership approachability. Clarity of guidance was raised by other participants as key to facilitating adherence; for example, RE (hospital B) noted that they would adhere if "it [the guidance] was clear cut. . . if it was very linear and easy to understand", and AE (hospital B) mentioned that they felt adherence was only possible because "it [the guidance] was clear". Thus, the approachability of leadership was not only seen to have facilitated positive perceptions of leaders, but also enabled staff to adhere through clarifying guidance.

Finally, leaders who facilitated two-way conversations were seen to be invested in what was best for frontline staff, particularly when the benefits and goals of the group were emphasised:

> We all knew that we were going into very uncertain times and nobody really knew the answer, so it was very much, you know [leadership saying] 'if you guys think that you have an idea for a better way for us to be doing this, you know, no idea is a bad idea'. We're all

just trying to look after each other as best we can here and so, you know, you did feel comfortable raising ideas or saying 'actually, I didn't feel safe in this situation'

(Extract 11, SA, hospital A)

Here, SA (extract 11), a frontline member of staff, views leaders as demonstrating ingroup membership by engaging in two-way conversations that emphasised shared goals (i.e., 'look after each other') with staff and therefore inviting leaders and staff to be on an equal level. By leaders being seen to care about the group and wanting to keep each other safe, they define what it means to be a good (and ideal) group member [14]. Thus, SA recognises this leader as representing prototypicality. In addition, leaders who engaged in these two-way conversations were seen to be working *with* staff and *for* staff to achieve shared group goals (to keep each other safe). By leaders acting for the group through engaging in conversation and co-production with frontline staff, they are positioned as identity advancing leaders.

**Engaged leadership who act upon staff's concerns.** Positive appraisals of leadership are also extended to leaders who are seen to act upon the concerns raised by staff:

The very first eye protection that we had, if you've never seen it, it was essentially like a cellophane sheet with a bit of coat hanger through it. And they fell off, and they didn't work, and everyone complained about them. And I think we all had faith that the person that was giving us, immediately giving us the rules, was also the person that was fighting our corner to the chain of command. And I think that's, yeah I think that would've persisted if there were any other things in that particular environment

(Extract 12, AE, hospital B)

AE (extract 12) describes an example where PPE inadequacy was raised as a concern to leadership, where leaders were viewed as demonstrating group advancing behaviours by acting upon these concerns on behalf of the group. AE raises an assumption that issues with inadequate PPE are a challenge faced by everyone and provides an example of the barriers to support their point. AE also poses that all frontline staff look to leadership to resolve these issues but AE assumes a conflict is required with higher-up management to resolve issues. AE implies a hierarchy and top-down approach to leadership which AE views as unhelpful for facilitating adherence. In contrast, AE positively views those providing frontline staff with guidance as those who would challenge upper management on their behalf. By demonstrating identity advancing behaviours of engaging in a difficult challenge on behalf of the group, these leaders were seen to facilitate feelings of trust that they would act for the group again in future, as seen within prior literature [21].

Leaders who were seen to not act on the concerns of staff were seen as dismissive and limited staff's ability to adhere to COVID-19 safety guidance. LY (extract 13), who worked within hospital office spaces rather than a patient facing role, discussed an example where their line manager did not promote working from home:

So we had a number of discussions and they were still pushing for me coming into the office for one day to send emails just because everyone was doing it [. . .] that's one of the examples when I felt that they were not listening

(Extract 13, LY, hospital A)

In this example, LY views leadership as failing to act on their concerns and actively dismissing them by not listening or engaging with the concerns raised by staff, and justifying a lack of

action by attributing it to what everyone else in the hospital was doing. A similar experience was discussed by NE (hospital B), who found that their line manager did not encourage working from home and instead was seen to perceive this "as an inconvenience" where "there's no real, you know, conversation at all" regarding the concerns being raised by office staff. As a result, these leaders are perceived as non-identity advancing leaders as they are seen to be failing to act for the group by not encouraging or facilitating working from home (which was the Scottish Government's guidance at the time [39]). These examples of leadership failing to act on staff's concerns, therefore, limited staff's ability to adhere to COVID-19 safety guidance.

In comparison, leadership who acted for the group by addressing the concerns raised by staff were both viewed positively and facilitated adherence to COVID-19 safety guidance as exemplified by SA (extract 14):

> They were very useful for being able to sort of, even if they didn't have the answers in there they would research into the trust to see if there's any other kind of cleaning products that we could use that would be a lot safer on the equipment and being able to get that in for us instead. And that was really helpful
>
> (Extract 14, SA, hospital A)

Here, SA demonstrates that leaders who are seen to be acting upon group concerns are appraised positively, where these engaged leaders do not leave staff feeling dismissed compared to leaders who are seen to be ignoring the views of the group as discussed previously. SA highlights an example where staff were unable to clean the equipment with the existing cleaning products as they would cause damage. By acting upon the concerns raised by staff regarding this, leadership found a solution that facilitated staff following the COVID-19 guidance. Thus, this highlights the importance of leadership acting upon group concerns (demonstrating identity advancement) not only for the positive appraisal of leadership, but also for facilitating the ability for staff to adhere to COVID-19 safety guidance.

## Discussion

This study explored the role of identity leadership and its implications for adherence to COVID-19 safety measures in hospital settings. Two over-arching themes were generated: leadership presence and approachable leadership who act on group concerns. Together, these themes demonstrate how present and approachable leadership act as antecedents to seeing leaders as being part of the same group and enabling leaders to engage in identity advancement actions.

### Present and approachable leadership as precursors to ingroup membership

In line with prior literature on leadership prototypicality (e.g., [19, 21, 22]), leaders who were seen to be present and participate in prototypical frontline tasks were viewed as sharing a sense of commonality with staff, which facilitated trust in their leadership. In stark comparison, non-present and non-approachable leaders were seen to create feelings of abandonment and a lack of support. Leaders' presence and approachability was seen to facilitate understanding of staff's experiences and insights into the challenges and concerns being faced by frontline staff, which contributed to viewing leaders as being part of the same team (as seen in prior literature [38]) alongside creating positive appraisals of leadership. Being present and engaging in discussions with staff also allowed them to craft a sense of commonality through shared challenges and goals (e.g., keeping each other safe).

Although broader leadership literature highlights the importance of approachability for positive appraisals of leadership and their role in promoting effective working [40, 41], the role of approachability within the identity leadership framework has yet to be considered. The present research bridges the gap between these two leadership literature bases and provides new insights into how presence and approachability can be precursors which facilitate the view of leaders as being part of the same group with staff, or at least sharing a commonality as a precursor to shared group membership.

## Present and approachable leadership facilitating staff goals

Throughout the interviews, present leaders were seen to facilitate adherence to the safety guidance by providing relevant and applicable guidance, particularly through having shared experiences with frontline staff. Although prior literature demonstrates leaders can promote safe behaviour via increased trust [24], discouraging unsafe behaviours [42], and providing insights into group appropriate behaviours [13], the present study provided alternative insights. Instead of highlighting leadership behaviour as an influence on their own behaviour or motivations to adhere, staff reported that leadership enabled them to perform actions they already wanted to do. Thus, instead of leadership *promoting* particular goals, present and approachable leaders in the present study were seen to *facilitate* staff goals by providing the practical guidance required.

Approachable leaders were viewed positively because they were seen to be on an equal level to frontline staff, create channels for sharing the information required for adherence to guidance, and act on staff concerns to facilitate staff's goals of acting safely. In some cases, this included leaders who worked closely with staff acting as a representative for their needs to other leaders higher in the leadership hierarchy who were viewed as being unaware of the challenges staff faced. This approachability was seen to provide insights to leaders regarding the challenges faced by staff, which leaders could choose to act upon (and therefore demonstrate identity advancement) by updating guidance or providing resources to facilitate adherence.

Leaders who were seen to demonstrate advancement by acting on the concerns raised by staff were appraised positively throughout the interviews. Consistent with prior literature [17], staff reported that identity advancing leaders facilitated greater feelings of trust and reassurance in leadership, and especially created feelings that leadership would continue to act upon the concerns of staff in future. However, the present research builds on this with a slightly different perspective: being present and approachable allowed an environment for identity advancement actions to occur.

## Implications for theory and practice

The present study provides new theoretical implications for the role of identity leadership within hospital settings. Leaders being viewed as present and approachable (such as by engaging in prototypical behaviours and listening to staff's concerns) provides avenues to leaders being seen as part of the same group and being able to engage in identity advancement actions on behalf of staff. These relational processes were key to effective leadership and occurred regardless of whether the leader was a line manager or in a higher management position: the engagement with staff and acting to help them achieve their goals facilitated positive appraisals and a sense of commonality with that leader.

Although prior literature suggests that prototypical leaders can promote safety (e.g., [13]), the present study found that leaders were instead seen to *facilitate* the ability for staff to adhere. This unexplored factor could potentially explain why recent quantitative research with hospitals found no significant association between prototypical leaders and self-reported adherence

to COVID-19 guidance [22]. Further, this also highlights the potential for the mechanisms by which identity leaders can encourage engagement in safe behaviours to vary across different contexts. For example, some contexts may rely on leaders demonstrating safe behaviours to facilitate followership, or simply facilitating the ability for group members to act safely because it is what they already want to do. Importantly, the findings suggest that leaders should engage with staff to understand their situation and barriers to achieving the desired goals, in order to build positive relations and work together with staff to achieve solutions. However, future research is required to explore the nuance and potential context dependency of the role of identity leadership in guiding safe behaviour, particularly beyond the current COVID-19 context.

The present research also provides routes for practical implications. For example, current NHS leadership guidance promotes concepts overlapping with identity leadership principles [7, 9], but training should be updated to recognise areas for improvement highlighted in the present research. For example, leadership presence on the wards, alongside engagement in frontline tasks, should be encouraged to promote perceptions of leaders as prototypical and to facilitate the creation of applicable guidance. This is particularly important for levels of management typically not present on the wards, where a more distributed approach to leadership could be adopted, i.e., those providing guidance and instructions to frontline staff should ensure presence and engagement to facilitate positive relations between staff and leaders and positive attitudes towards the guidance. In addition, the value of engaging in the co-production of safety guidance with frontline staff should be emphasised, where this can assist in providing practical and applicable guidance that staff can adhere to, including within non-COVID contexts.

Further, the present research highlights practical implications when considering moving to hybrid models of working. Leaders' presence was discussed in the present study as relating to understanding the environment, experiences, and challenges faced by frontline staff. Moving to hybrid models of working, particularly within healthcare settings where frontline staff remain in-person, may mean that these relational connections and shared experiences between leaders and staff (which are antecedents of shared social identity) are missing. This is not to suggest that hybrid models of leadership are necessarily negative, but it does highlight the need for strategy development to ensure that leaders are able to develop an understanding of the experiences and working environments of their staff.

## Strengths, limitations, and future research avenues

The interviews provide unique and nuanced insights that highlight routes for advancing our theoretical understanding of the role of identity leaders in facilitating safe behaviour, particularly within healthcare settings. The research also contributes insights to quantitative research conducted in the same hospitals throughout the period of this project which explored the links between staff relations with leadership and measures of self-reported adherence [22], indicating that the present findings could generalise beyond the evaluated sample.

However, the present study has limitations. For example, the present study did not evaluate the role of staff's social identities on appraisals of leadership, particularly the role of identification as team members, given that prior literature highlights the role of salient group identities on appraisals of identity leaders [43, 44]. Thus, future research should evaluate the potential for varying levels of social identification as a healthcare worker, alongside staff identification as a member of their team, to influence the role of identity leadership principles on (a) appraisals of leadership and (b) adherence to safety guidance beyond the COVID-19 context.

Leaders being seen to care for staff was raised within the interviews, where this was appraised positively and related to staff adherence to the safety guidance. However, future research should explore how caring leadership may allow leaders to understand and address the needs of staff, and how this contributes to positive relations because staff are being understood and helped.

In addition, the months in which the present research took place (October–December 2021) provide potential limitations. Prior research suggests that followers look to leaders more during periods of uncertainty, where non-prototypical leaders gain additional support during these times [45]. The present research did not evaluate the potential influence of staff uncertainty on appraisals of leadership or on the role of leadership in promoting safe behaviour. Future research should therefore consider the influence of varying levels of uncertainty, particularly considering how the role of identity leaders may vary with changes in staff uncertainty over time. Further, our interviews were relatively short since NHS staff availability was limited and precious during COVID-19. Greater availability of staff for interviews would have provided more opportunities for in-depth discussion of staff relations with leadership. However, we believe our data has strong information power [46] given staff's rich knowledge and experience of leadership during the pandemic.

It is important to recognise that the methodology used in the present study limits our ability to draw conclusions and make definitive recommendations for the entire NHS. Indeed, our epistemological approach roots our findings in the subjective experiences of the staff interviewed, rather than providing an objective evaluation of what constitutes effective leadership. Further, our research is focused on staff's own experiences and perspectives, in line with the experiential approach taken, and this operates within the context of COVID-19 when both frontline staff and leadership were exceptionally overstretched and navigating new guidance and procedures. However, this does not negate the key contribution of the present research to understanding the complex relationships between frontline staff and leaders and the implications these have for staff's views on the safety of their working conditions.

Relatedly, the present research does not obtain objective measures of adherence within hospitals. Behavioural data could be evaluated in future research to explore staff adherence levels, particularly exploring whether these vary following (a) demonstrations of safe behaviour by identity leaders and (b) demonstrations of leaders enacting group advancing behaviours. This would provide quantitative insights into how effectively identity leaders can influence engagement in safety guidance.

Further, the present analysis relied on spontaneous references to group or team membership and the use of togetherness language given the limited questions within the interview schedule related to relationships with leadership due to the rapid response nature of the project. Future research should therefore ensure questions related to leadership group membership are key within the interview schedule to reduce reliance on spontaneous mentions of group membership.

Finally, there were limited questions within the present interview schedule that explored leadership since this study was part of a larger project. This limited the range of example quotes available for the present analysis. Going forward, research could look in more depth at the themes generated in the present research, including evaluating these across a new sample of NHS staff. This could include evaluating dimensions of identity leadership not considered in the present research, potentially considering how leaders can create and encourage a sense of 'us' between staff (identity entrepreneurship and identity impresarioship [9]) and whether this can promote safe behaviour.

## Conclusion

The present study explored identity prototypicality and identity advancement within interviews across two NHS Scotland hospitals. Through Reflexive Thematic Analysis, two overarching themes were generated: leadership presence and approachable leadership who act on group concerns. Present and approachable leadership were found to act as antecedents to either seeing leaders as part of the same group or at least sharing commonality with frontline staff. Additionally, present and approachable leadership enabled staff to perform the actions/goals they already wanted to engage in, such as adhering to the COVID-19 safety guidance, where leaders' presence and approachability created an environment for identity advancement actions to occur. Although rooted within the methodological and philosophical assumptions of the present research, these findings provide routes for theoretical advancements in identity leadership and avenues for future research. Furthermore, they provide opportunities for practical interventions, such as developing leadership training to ensure strong relationships between leaders and frontline staff, alongside promoting adherence to general safety guidance beyond COVID-19.

## Supporting information

**S1 File.**
(DOCX)

## Acknowledgments

Our sincere thanks go to all the healthcare workers who participated in our research and spent their valuable time guiding the progression of the overall project. We also thank Lesley Shepherd, Susie Dodd, Lisa Powell, and Steve Reicher for their expert input into the project. Thank you to the research team, Kirsty Wiseman-Gregg, Jean Skelton, Eve Stanley, Chiara Addison and Gareth Clegg, without whom the project would not be possible.

## Author Contributions

**Conceptualization:** Kayleigh Smith, Anne Templeton.

**Data curation:** Kayleigh Smith.

**Formal analysis:** Kayleigh Smith, Anne Templeton.

**Funding acquisition:** Anne Templeton.

**Methodology:** Kayleigh Smith, Anne Templeton.

**Supervision:** Anne Templeton.

**Writing – original draft:** Kayleigh Smith.

**Writing – review & editing:** Kayleigh Smith, Anne Templeton.

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
