## [Decision Letter · Decision Letter 0]

2 May 2023

PONE-D-23-07018Identity leadership and adherence to COVID-19 safety measures in hospital settingsPLOS ONE

Dear Dr. Templeton,

Thank you for submitting your manuscript to PLOS ONE. After careful consideration, we feel that it has merit but does not fully meet PLOS ONE’s publication criteria as it currently stands. Therefore, we invite you to submit a revised version of the manuscript that addresses the points raised during the review process.

I and the two reviewers all felt that this was a largely well conducted, interesting, and well reported study that comes close to meeting the requirements for publication in this journal. Nonetheless, it will require further revision to meet our requirements for clarity of  reporting, explanation and argument.

I should like you to revise your paper to address the concerns raised by the reviewers. Whilst you should respond to all points raised by the reviewers, it will be particularly important to address the following: -

1. Strengthen your discussion of the links between the data and your conclusions about identity.

2. Provide an explanation for the brevity of interviews and why they provide sufficient data to address the research question.

3. This journal does not provide proof reading, so it will be particulary important to ensure you have carefully proof read the paper before resubmission, correcting all errors of grammar and any typos.

We look forward to receiving your revised manuscript.

Kind regards,

Prof. Mark Fenton-O'Creevy

Academic Editor

PLOS ONE

Journal Requirements:

"This research is funded by the Chief Nursing Officer Directorate of Scottish Government for the project ‘Identifying barriers and avenues to safe behaviours in high-risk workplaces’."              

"N/A"

Reviewers' comments:

Reviewer's Responses to Questions

**Comments to the Author**

1. Is the manuscript technically sound, and do the data support the conclusions?

Reviewer #1: Yes

Reviewer #2: Partly

2. Has the statistical analysis been performed appropriately and rigorously? 

Reviewer #1: N/A

Reviewer #2: N/A

3. Have the authors made all data underlying the findings in their manuscript fully available?

Reviewer #1: Yes

Reviewer #2: Yes

4. Is the manuscript presented in an intelligible fashion and written in standard English?

Reviewer #1: Yes

Reviewer #2: Yes

5. Review Comments to the Author

Reviewer #1: Overall this is a neat, well designed and written up study, of limited but important scope. I see no major shortcomings of the study. However, I have highlighted a methodological query concerning the length of interviews and also the need to acknowledge the shortcomings of the model of leadership adopted as a conceptual framework.

Introduction:

The introduction is clearly written. The authors make clear the topic to be addressed and they underline its importance both to the NHS and society as a whole. The nature of the problem as a leadership problem is clearly and convincingly stated – i.e. that this is an issue regarding shaping meaning making.

Theory:

A clear presentation of a social identity view of leadership was offered. I perhaps would have liked to have seen how the theory allows for personal identity can affect group behaviour – as this is surely not only restricted to social identity. A small elaboration on the four dimensions of Steffens et al’s framework would have been useful in order to better understand why two dimensions were considered relevant and the other two not.

The presentation of identity advancement and identity prototypicality was clear and straightforward. I had some questions related to who was being approached in the study as constituting the group – the boundaries placed around the ‘group’.

Methodology:

The methodology was clear and suitable. I had a question as to the length of interviews and was surprised that they did not seem on the whole to last very long. I had assumed that research participants would have a lot of stories to tell of their time working through the pandemic. I wondered therefore whether a narrative approach to interviewing might have elicited more data. This is not to ask the authors to revisit their approach but to request an explanation for why they think interviews tended to be briefer, and perhaps also to offer some reflections in the concluding sections of the paper on limitations of the approach adopted.

It would have been useful in addition to have more detail on what the six-phase structure of analysis consisted of.

Findings:

In terms of findings, I was taken by the finding that being physically co-present with staff was important to perceptions of effective leadership. This is a relevant finding as the world moves to more hybrid models of working. I wondered if this could be highlighted a bit more in discussion, particularly acknowledging literature on relational and embodied approaches to leadership, which seem salient for future research to better understand this phenomenon. The quote on p.13 about embodied experiences was significant in this regard, suggesting the power of being there – experiencing the heat and physical discomfort and of this taking on a power when it is a shared, embodied experience.

The finding on the need for a mediator with on-ground experience was also interesting, perhaps and argument for a more distributed approach to leadership, which could again be highlighted and discussed.

There also seems to be a connection between approachability and care/caring leadership, which could be highlighted as an area for further exploration. There is of course a tension here between caring leadership that adapts to need and the competing need for consistency from management.

Discussion:

The discussion section was thorough and substantial. I liked the way in which the authors signalled the limitations of the time period in which the research was conducted, indicating that findings may have been different in a different time period – i.e. one of more or less crisis.

However, a flaw in the whole paper to me seems to be the limitations of the model of leadership adopted as a conceptual framework. While in and of itself it did its job, there are limitations that could more strongly be explored in the discussion. This was most apparent when the authors discuss the facilitative role of people in formal positions of leadership. To me this finding was not particularly surprising because I tend to view leadership as a process rather than as equated to a formal organisational position. Ultimately the authors do equate leadership to a position of formal authority. I recommend some further critical reflection on the conceptual framework. To me, the issues discussed at the end of the paper reveal that leadership is a shared process whereby leadership can come from the front lines as well as from on high – it is relational. In other words, adopting an alternative view of leadership would help to develop insight that leadership in crisis conditions needs to be re-envisaged in relational terms. I am not requesting that the authors adopt a different conceptual framework, only that they acknowledge that alternative views of leadership could enrich understanding of their findings and similar studies in future research.

Reviewer #2: This paper reports an interview study with 25 hospital staff that explores their relationship with/ view of their management and the relationship of this to safety practices, in particular their adherence to Covid safety guidelines. The study is situated appropriately in an up to date literature review. The rationale is logical. The study is also well designed and conducted and with a very thorough and comprehensive account of methodology. The analysis competent and mostly convincing. The study will be of interest to social identity researchers and those who work in organizations and health services.

This is a valuable study that enables the development of a hypothesis (that lack of shared social identity is a key to problems in hospitals and that addressing shared social identity is a solution). But it doesn’t address that hypothesis itself, because the data presented say little about social identity. The interview schedule does not ask people about group memberships etc. and maybe this is why many of the extracts presented do not actually refer to it. If the interview schedule doesn’t ask people about the topic, then the researchers are reliant on spontaneous references to it, in the form of ‘we’-talk, for example. But there was only a small amount of this. Overall the antecedents (e.g. shared experience) and consequences (e.g., support) of shared social identity are evident in this analysis, but social identity itself is missing. Unless the evidence exists elsewhere in the dataset, the framing should be changed accordingly.

Specific points and suggestions

Is it necessary to put the indented quotes in italics? This is not very accessible. (Also indented quotes don’t need quote marks.)

p. 2 line 65 Is ‘demonstrates’ the correct word here?

p. 3 line 86 Is ‘conceives’ the right word here?

p. 3 line 90 I’m not sure what ‘achieving influence at the group’ means. Against the group?

p. 3 line 103 Is this OSF site the date for the present study or a different one? It’s not clear from the wording. I’m confused by the status of the ‘conversations’ mentioned here.

Methods

Why were nine transcripts selected separately? I’m not clear why these had more in depth discussion of social relations.

14 mins is very short for an interview. Did you manage to get through all the questions?

P 9 line 234 I’m not sure what was actually done when you say interview responses were triangulated against some survey data.

P. 9 lines 244-246 ‘Across the interviews, staff highlighted the importance of leadership being present to perceiving those leaders as prototypical members, which was related to positive appraisals of leadership alongside facilitating adherence to guidance.’

Sentence is quite difficult to process.

It might be useful to number the quotes/ extracts.

The first extract (RE hospital B) says the leader has been present but I can’t see any praise there as the first sentence of commentary (lines 255-256) states.

Quote on p. 11 lines 269-274. I can see that this could be explained by the interviewee seeing the manager as ingroup, but that kind of claim doesn’t seem to be in the extract itself. A bit more evidence is needed here. The commentary para ends with a reference to the literature on trust and prototypicality but again the quote doesn’t refer to trust.

Lines 302-3-4: ‘Thus, non-present

leadership who are seen to lack a shared understanding with staff could limit the ability for

staff to adhere to COVID-19 safety guidance.’

It would be better to demonstrate this relationship in the data rather than infer it (such inferences would be better in the Discussion where the implications of the research can be drawn out).

Line 321 ‘contributing to categorising distant leaders as outgroup’

In the extract, the interviewee says that because senior management worked from home they could not understand the problems encountered by frontline staff. Again I would have liked to see more direct evidence of interviewees categorizing people as ingroup or outgroup, either by asking the interviewee (e.g. ‘did you feel part of the same team’, ‘did you feel unity with them?’) or spontaneous references to being part of the same group or ‘we’-talk.

Line 326 NO, hospital B

Again this extract talks about lack of shared experience being a problem but the commentary states that this then leads to no shared group membership. I can’t see the latter in the extract.

Line 360

‘Further, by highlighting that distant leadership lacked this shared

experience, AE continues to categorise distant, non-present, and non-shared experience

leaders as outgroup.’

But where? This is a reasonable inference but it’s not in the data.

Line 434: SA Hospital A.

This extract is much more clearly an example including evidence of social identity, in the form of the ‘we’-talk.

6. PLOS authors have the option to publish the peer review history of their article (what does this mean?). If published, this will include your full peer review and any attached files.

Reviewer #1: No

Reviewer #2: No

---

## [Author Response · Author response to Decision Letter 0]

4 Sep 2023

Dear Editor and reviewers, 

Thank you for the time you have taken to provide us with your valuable feedback. The comments were insightful and have substantially improved our manuscript. We have made multiple changes following your feedback which we list below. 

Introduction 

Reviewer 1 requested more information on “how the theory (social identity view of leadership) allows for personal identity can affect group behaviour – as this is surely not only restricted to social identity”.

We thank the reviewer for this fair point. However, we feel that focusing on personal identities is beyond the scope of this particular paper. The interviews focused on team relations and adherence to the guidance. At times, personal factors such as not wanting to take disease home to loved ones came up, but these were not related to our main research questions on views of identity leadership.

Reviewer 1 requested “a small elaboration on the four dimensions of Steffens et al’s framework” to “better understand why two dimensions were considered relevant and the other two not”. 

We have expanded this description to explain the four dimensions of Steffens et al’s model to more clearly demonstrate why we selected two dimensions (prototypicality and advancement) as most relevant to explore (see lines 101-107). In this section, we also clarified how the early conversations with NHS staff highlighted the relevance of these two dimensions.

Reviewer 1 requested clarity on “who was being approached in this study as constituting the group”.

The group is subjective, and the participants themselves brought up who was relevant for their group. We appreciate that this was not clear so we have added a clarification within the manuscript when discussing the interview schedule in the methods section that staff were able to create their own categorisation of who was part of their group/team (see lines 218-220). 

Reviewer 2 made suggestions about specific line numbers which we addressed: 

p. 2 line 65: Is ‘demonstrates’ the correct word here? 

We have changed ‘demonstrates’ to ‘proposes’ (see line 66). 

p. 3 line 86: Is ‘conceives’ the right word here? 

We have changed ‘conceives’ to ‘highlights’ (see line 87).

p. 3 line 90: I’m not sure what ‘achieving influence at the group’ means. Against the group? 

Achieving influence at the group is not necessarily against the group but not within the best interests of the group (e.g., by engaging in more authoritarian styles of leadership which do not engage through the group but instead try to guide change by directing at those they are trying to lead). 

p. 3 line 103: Is this OSF site the date for the present study or a different one? It’s not clear from the wording. I’m confused by the status of the ‘conversations’ mentioned here.” 

The OSF site is a project for the present study, where the final report from the broader project has also been made available for this paper. These initial conversations were part of the broader project before the interviews that provides the data for this paper. We have clarified that these were initial discussions with staff (see lines 105-108).

Methods

Reviewer 1 and Reviewer 2 both highlighted where some interviews were particularly short and questioned why the interviews were not as long as we would typically expect.

We recognise the interview time varied, which was often due to limited staff availability during the height of the pandemic, for example staff taking time on their breaks to talk to us and having to quickly stop the interview to turn to work. We have added a clarification into the participants section to highlight this context (see lines 204-206), and have also added more about this within the limitations section of the discussion (see lines 658-662). 

Reviewer 1 requested “more detail on what the six-phase structure of analysis consisted of”. 

We have added an additional section providing more detail about the six-phase structure of Thematic Analysis set out by Braun and Clarke (see lines 227-233). 

Reviewer 2 requested clarification as to why nine transcripts were selected separately. 

We conducted a familiarisation phase on all transcripts prior to in-depth analysis. We then selected nine transcripts due to them providing in-depth, rich data for initial coding and theme creation. After initial in-depth analysis on these nine transcripts, we applied the coding to the remaining transcripts. We have now clarified this in the manuscript (see lines 197-199). 

Reviewer 2 highlighted that “14 minutes is very short for an interview. Did you manage to get through all the questions?”

We were able to get through all the priority questions marked in the interview schedule in the 14 minute interview as we knew time may be short with participants. We have added clarification in the methods section to show that we had priority questions in the interview schedule in case our participants did not have much time available for the interview (see lines 212-213). 

Reviewer 2 requested clarification when we stated that interview responses were triangulated against survey data. 

We have removed the section where we suggest that our findings were triangulated against survey data. Although we do reference a published article from this project, we had originally aimed to publish more of the quantitative data prior to this publication to therefore triangulate the qualitative findings to the quantitative survey data. However, as this is not yet available, we are unable to refer to other survey data at this time and have removed this section for clarity. 

Analysis 

Reviewer 2 asked for the first sentence of the analysis to be reworded as the sentence was quite difficult to process. 

We have reworked this sentence to improve clarity and readability (see lines 257 - 258). 

Reviewer 2 requested format changes to the quotes, and that we include extract numbers. 

We have removed the quote marks and italics on the quotes to improve accessibility and have now included extract numbers. This can be seen throughout the analysis section of the manuscript. Thank you for raising this so that we could improve the accessibility of the article.

Reviewer 2 stated that “the first extract (RE, hospital B) says the leader has been present, but I can’t see any praise there as the first sentence of commentary states”. 

We recognise that the praise is not clear from the extract itself and required further context. The response from RE regarding presence was following a question regarding their attitudes towards line management, and whether it was a feeling of togetherness or separation with their managers. In response, RE mentioned that they found no change from pre-covid, including their leader's presence and a lack of reluctance to get stuck in, therefore praising the presence of leadership. We have added this clarification into the manuscript (see lines 274-275). We have also reframed this section of the analysis to highlight the positive appraisal from leaders being present and engaging in the same tasks as frontline staff (see lines 277-281). 

Reviewer 2 highlighted how our arguments were less about shared social identity, but antecedents and consequences to shared social identity, and how our analysis required some reframing to address this. 

We thank reviewer 2 for their insight and have quite extensively reframed the analysis accordingly. Please find responses to specific comments below: 

Reviewer 2 highlighted that the quote on p11 (extract 2) that they “can see this could be explained by the interviewee seeing the manager as ingroup, but that kind of claim doesn’t seem to be in the extract itself. A bit more evidence is needed here. The commentary para ends with a reference to the literature on trust and prototypicality but again the quote doesn’t refer to trust”

We have reframed this section of the analysis, where we now highlight that leaders’ presence and engagement in the same tasks as frontline staff provides antecedents to feeling part of the same group as leadership because these actions signal leaders and staff being equal or being prototypical (see lines 283-284, line 295, and lines 298-300). 

Reviewer 2 highlighted that the analysis of extract 3 that our statement that “non-present leadership who are seen to lack a shared understanding with staff could limit the ability for staff to adhere to COVID-19 safety guidance” was an inference, where “it would be better to demonstrate this relationship in the data rather than infer it (such inferences would be better in the Discussion where the implications of the research can be drawn out)”

We have removed the inference of leadership limiting the ability for staff to adhere and have kept this just in the discussion section. 

Reviewer 2 highlighted that they would like “to see more direct evidence of interviewees categorising people as ingroup or outgroup, either by asking the interviewee (e.g., ‘did you feel part of the same team’, ‘did you feel unity with them?’) or spontaneous references to being part of the same group or ‘we’-talk”

We have reframed this section of the analysis to highlight that the non-presence of leadership is seen to leave leaders unable to understand the challenges being faced by frontline staff, where this lack of understanding from leadership creates a separation between frontline staff and leadership, in turn contributing to categorising these non-present leaders as ‘other’. We have also now highlighted that the non-presence of leadership and lack of understanding of frontline challenges creates a lack of relational connection between leaders and staff, which demonstrates a lack of antecedents to shared social identity through leadership distance (see lines 308-322). We have also added a section within our limitations highlighting that we did rely on spontaneous mentions of group or team identity, where we allowed staff to create their own definition of who they considered part of their group/team given that this is dependent on the ward and staff structure (see lines 679-684 for limitations section, and lines 218-220 for discussion of staff crafting their own definitions of who was part of their group/team given variability in ward and staff structures). 

Reviewer 2 highlighted for extract 5 that “this extract talks about lack of shared experience being a problem but the commentary states that this then leads to no shared group membership. I can’t see the latter in the extract”

We have reframed the analysis to highlight instead that the lack of shared experience renders the leaders in the office block as ‘other’ to frontline staff on the ward, where prior literature demonstrates the role of shared experiences in recognising others as part of the same social group (see lines 357-360). 

Reviewer 2 highlighted for extract 6 that our statement “Further, by highlighting that distant leadership lacked this shared experience, AE continues to categorise distant, non-present, and non-shared experience leaders as outgroup” was “a reasonable inference but it’s not in the data”.

We have reframed this section of the analysis to instead highlight that AE’s comments demonstrate how antecedents to feeling like part of the same group as leadership are missing when these leaders are not present and do not share the same experiences as frontline staff (see lines 381-386). 

Discussion

Reviewer 1 highlighted the finding that “being physically co-present with staff was important to perceptions of effective leadership” which is a “relevant finding as the world moves to more hybrid models of working” and requested for this to “be highlighted a bit more in discussion, particularly acknowledging literature on relational and embodied approaches to leadership which seem salient for future research to better understand this phenomena”.

We don’t feel we have enough evidence related to embodied approaches to leadership to discuss this in depth. However, we have added further discussion of the importance of considering the role of physical presence on effective leadership. We have discussed the importance of presence on creating an understanding of staff’s environment, experiences, and challenges faced in their workplace. We have highlighted that this is not to suggest that hybrid models of working are necessarily negative, but that strategy development is required to ensure leadership are able to develop this understanding of the staff environment/experience (see lines 620-628). 

Reviewer 1 highlighted “the finding on the need for a mediator with on-ground experience was also interesting”, noting that this was “perhaps an argument for a more distributed approach to leadership which could be highlighted and discussed”. 

We have highlighted within the practical implications section that a more distributed approach to leadership could be adopted, i.e., where those providing guidance and instruction to frontline staff could ensure their presence and engagement in order to facilitate both positive attitudes towards these leaders but to also promote positive attitudes towards the guidance they are providing (see lines 614 -617). 

We have also highlighted the importance of leaders engaging with staff in order to understand the situation and barriers they are facing to build positive relations with staff to achieve solutions to these (see lines 602-605). 

Reviewer 1 highlighted that “there also seems to be a connection between approachability and care/caring leadership, which could be highlighted as an area for further exploration”

We have added a discussion of how leaders being seen to care for staff was raised, which was appraised positively by staff and related to staff adherence to safety guidance. We have highlighted that this was mentioned only a few times, where we need future research to further explore the implications of caring leadership on the opportunities for leaders to (a) understand and (b) address the needs of staff (see lines 645-650). 

Reviewer 1 requested that we provide an explanation for why we felt the interviews tended to be brief, and “offer some reflections in the concluding sections of the paper on the limitations of the approach adopted” 

We have included an explanation for why some interviews were brief, which was primarily due to NHS staff availability being short and precious during COVID-19, where greater staff availability with staff would have provided more opportunity for in-depth discussions about staff relationships with leadership (see lines 658-662). 

Reviewer 1 recommended “some further critical reflection on the conceptual framework. To me, the issues discussed at the end of the paper reveal that leadership is a shared process whereby leadership can come from the front lines as well as from on high – it is relational. In other words, adopting an alternative view of leadership would help to develop insight that leadership in crisis conditions needs to be re-envisaged in relational terms” 

We have added further reflection and highlight now that the relational processes of presence and approachability were key to effective leadership and occurred regardless of formal leadership position (see lines 587-593).

---

## [Editor Report · Decision Letter 1]

4 Oct 2023

Identity leadership and adherence to COVID-19 safety measures in hospital settings

PONE-D-23-07018R1

Dear Dr. Templeton,

We’re pleased to inform you that your manuscript has been judged scientifically suitable for publication and will be formally accepted for publication once it meets all outstanding technical requirements.

Kind regards,

Mark Fenton-O'Creevy, PhD

Academic Editor

PLOS ONE

Additional Editor Comments (optional):

Thank you for your careful responses to reviewer concerns. I am delighted to now move this interesting and useful paper forward in the publication process. Please note that the journal does not provide a proof reading process so you should carefully proofread your manuscript, once technical checks are complete.

Reviewers' comments:

Not sent out for futher review.

---

## [Editor Report · Acceptance letter]

23 Oct 2023

PONE-D-23-07018R1 

Identity Leadership and Adherence to COVID-19 Safety Guidance in Hospital Settings 

Dear Dr. Templeton:

I'm pleased to inform you that your manuscript has been deemed suitable for publication in PLOS ONE. Congratulations! Your manuscript is now with our production department. 

Kind regards, 

on behalf of

Professor Mark Fenton-O'Creevy 

Academic Editor

PLOS ONE